# Aberrant Dopamine System Function in the Ferrous Amyloid Buthionine (FAB) Rat Model of Alzheimer’s Disease

**DOI:** 10.3390/ijms24087196

**Published:** 2023-04-13

**Authors:** Stephanie M. Perez, Angela M. Boley, Alexandra M. McCoy, Daniel J. Lodge

**Affiliations:** 1Department of Pharmacology and Center for Biomedical Neuroscience, UT Health San Antonio, San Antonio, TX 78229, USA; boley@uthscsa.edu (A.M.B.); lodged@uthscsa.edu (D.J.L.); 2South Texas Veterans Health Care System, Audie L. Murphy Division, San Antonio, TX 78229, USA

**Keywords:** Alzheimer’s disease, psychosis, dopamine, FAB, ventral hippocampus, in vivo electrophysiology

## Abstract

Antipsychotics increase the risk of death in elderly patients with Alzheimer’s disease (AD). Thus, there is an immediate need for novel therapies to treat comorbid psychosis in AD. Psychosis has been attributed to a dysregulation of the dopamine system and is associated with aberrant regulation by the hippocampus. Given that the hippocampus is a key site of pathology in AD, we posit that aberrant regulation of the dopamine system may contribute to comorbid psychosis in AD. A ferrous amyloid buthionine (FAB) rodent model was used to model a sporadic form of AD. FAB rats displayed functional hippocampal alterations, which were accompanied by decreases in spontaneous, low-frequency oscillations and increases in the firing rates of putative pyramidal neurons. Additionally, FAB rats exhibited increases in dopamine neuron population activity and augmented responses to the locomotor-inducing effects of MK-801, as is consistent with rodent models of psychosis-like symptomatology. Further, working memory deficits in the Y-maze, consistent with an AD-like phenotype, were observed in FAB rats. These data suggest that the aberrant hippocampal activity observed in AD may contribute to dopamine-dependent psychosis, and that the FAB model may be useful for the investigation of comorbid psychosis related to AD. Understanding the pathophysiology that leads to comorbid psychosis in AD will ultimately lead to the discovery of novel targets for the treatment of this disease.

## 1. Introduction

Approximately 6.2 million Americans are living with Alzheimer’s disease (AD), a debilitating and progressive neurodegenerative disorder. One often overlooked and prominent comorbid feature of AD involves psychotic symptoms (hallucinations and delusions). Psychosis is present in approximately 50% of individuals with AD [1,2], and has been linked to more rapid cognitive deterioration as well as an increased incidence of other psychiatric and behavioral disturbances [3,4,5,6,7]. Further, a black box warning was issued by the Food and Drug Administration (FDA) for all first- and second-generation antipsychotic medications, as they can induce cardiovascular and metabolic abnormalities that contribute to the increased risk of death in elderly dementia patients [8,9,10,11,12,13]. Thus, therapies to manage psychosis in AD are far from adequate, and gaining a better understanding of the pathophysiological alterations contributing to psychosis in AD will give rise to novel therapeutic targets for this disorder.

The hippocampus plays a key role in learning and memory, and is consistently implicated in AD, where progressive memory impairment is a key feature [14]. Structural and functional alterations in the hippocampus are central aspects of the pathophysiology of AD [15]. MRI studies in humans have reported that hippocampal alterations can be valuable predictors of the transition from mild cognitive impairments to AD [16,17,18]. Further, we have previously demonstrated that the ventral regions of the hippocampus (corresponding to the anterior in primates) play a fundamental role in the regulation of the dopamine system, and that aberrant regulation likely contributes to psychosis in models that display schizophrenia-like pathophysiology [19,20,21]. Interestingly, the anterior regions of the hippocampus are significantly affected in AD [22]. This is of particular importance as the anterior hippocampus innervates limbic regions, such as the prefrontal cortex, amygdala, and nucleus accumbens [23]. Thus, aberrant hippocampal activity not only leads to memory deficits, but also to aberrant control of the mesolimbic dopamine system, resulting in alterations in mood and executive function.

The neuropathology of AD is characterized by extracellular plaques which contain amyloid β (Aβ) peptide and neurofibrillary tangles with hyperphosphorylated tau. Few studies have demonstrated a relationship between AD pathology and its correlation with neuropsychiatric disorders; however, it has been suggested that AD pathologies may be associated with neuropsychiatric symptoms in the early stages of the disease [24,25,26,27,28]. Aβ is, indeed, produced in healthy brains, and is required for synaptic plasticity [29]; however, Aβ can dose-dependently affect learning and memory [30]. Further, there is significant literature suggesting that Aβ plaque deposits contribute to region-specific increases in hippocampal activity in AD. Specifically, Aβ causes impairments in inhibitory interneurons, resulting in a disinhibition of pyramidal neurons and network hyperexcitability [31,32,33]. While Aβ itself may directly contribute to pockets of hippocampal hyperactivity, it may also indirectly contribute by damaging interneurons [31,32,33,34]. Interneurons that express the calcium binding protein parvalbumin (PV) have been implicated in multiple psychiatric disorders, including AD, in which post mortem studies have confirmed a robust (over 50%) decrease in hippocampal parvalbumin [35,36,37,38]. Progressive hippocampal atrophy underlies the impaired episodic memory that is a hallmark of AD [22], and PV interneurons may be more susceptible to the oxidative damage present in AD because they exhibit fast-firing patterns and have high metabolic demand [39]. It is probable that the loss of PV in AD may also contribute to aberrant hippocampal regulation of the mesolimbic dopamine system, resulting in comorbid psychosis. Previous findings from our laboratory demonstrated a decrease in the expression of PV in rodent models of psychosis, which also displayed diminished oscillatory activity, aberrant hippocampal activity, and subsequent dysregulation of the dopamine system [19,40,41,42,43]. Specifically, ventral hippocampal (vHipp) knockdown of PV expression was sufficient to induce the aberrant dopamine signaling that is associated with psychosis [44].

Given that the hippocampus is a key site of pathology in AD, and that aberrant hippocampal regulation of the dopamine system contributes to psychosis in schizophrenia, we posit that the hippocampus may be a site of pathology contributing to comorbid psychosis in AD. Specifically, hippocampal hyperexcitability, caused by Aβ accumulation, underlies psychosis in AD via an interaction with the mesolimbic dopamine system. Herein, we used the ferrous amyloid buthionine (FAB)-treated rat to model AD-related pathologies that correspond to the sporadic form of the disease, representing approximately 95% of all cases [45]. While age is a contributing factor to AD, this model is experimentally advantageous as it rapidly recapitulates key pathological hallmarks of AD, including amyloid deposits, hyperphosphorylated tau protein, neuronal loss, and gliosis [45]. It also allows for alterations in cognitive function to be attributed specifically to AD-like pathology, rather than natural aging. We examined the basal activity of the vHipp and the dopamine neuron activity of the ventral tegmental area (VTA), as well as behaviors associated with AD and psychosis in both FAB and control rats. Understanding how alterations in basal vHipp activity can modulate dopamine neurons and the role each plays in AD pathology is vital to the development of novel pharmacological therapies to treat these devasting symptoms in the elderly population.

## 2. Results

### 2.1. Firing Rates of Putative Pyramidal Neurons in the vHipp Were Increased, While Coordinated Neuronal Activity Was Significantly Decreased in FAB Rats

Rodent models that display psychosis-related pathologies often exhibit hippocampal hyperactivity [40,41,44,46]. Consistently with these observations, FAB rats displayed a vHipp firing frequency (*n* = 87 neurons; 0.68 ± 0.06 Hz) that was significantly higher than that in control rats (*n* = 90 neurons; 0.51 ± 0.05 Hz; Mann–Whitney rank sum test; *t* = 8733.00; *p* = 0.004; Figure 1A). Further, recordings of spontaneous oscillatory activity in the vHipp (Figure 1B,C) were significantly reduced in FAB rats (*n* = 6–7 per group; two-way ANOVA; F_Strain(1,64)_ = 12.972; F_Frequecy(4,64)_ = 8.486; Holm–Sidak; *t* = 3.602; *p* < 0.001; Figure 1D). Specifically, FAB rats displayed significant decreases in slow-wave, high-amplitude frequencies in the delta (0–4 Hz; Holm–Sidak; *t* = 3.773; *p* < 0.001) and theta (4–8 Hz; Holm–Sidak; *t* = 2.502; *p* = 0.015) ranges (Figure 1D). No significant differences were observed in any other frequency ranges.

### 2.2. FAB Rats Displayed a Significant Increase in Dopamine Neuron Population Activity

Consistently with previous reports, control rats displayed a population activity of 1.03 ± 0.05 cells per track (*n* = 12 rats) [19,47]. Interestingly, a significant increase in dopamine neuron population activity was observed in FAB rats (*n* = 12 rats; 1.71 ± 0.14 cells per track; Mann–Whitney rank sum test; *t* = 93.50; *p* = 0.001; Figure 2A), as is similar to observations in rodent models used to study psychosis [19,48,49,50]. No differences were observed in the average firing rate (Figure 2B) or percent bursting (Figure 2C) between groups.

### 2.3. FAB Rats Displayed Augmented Locomotor Activity and Deficits in Working Memory

The acute administration of psychomotor stimulants elicits an exaggerated locomotor response in rodent models used to study psychosis [19,41,46]. Here, we used the N-methyl-D-aspartate (NMDA) receptor antagonist, MK-801, a psychotomimetic compound that produces locomotor-stimulating effects at subanesthetic doses. Interestingly, we demonstrated that both baseline (two-way ANOVA; F_Strain(1,101)_ = 30.085; Holm–Sidak; *t* = 5.485; *p* < 0.001) and MK-801-induced locomotor activity were elevated in FAB rats (*n* = 8 rats) compared to the controls (*n* = 9 rats; two-way ANOVA; F_Strain(1,152)_ = 32.797; Holm–Sidak *t* = 5.727; *p* < 0.001; Figure 3A).

Age-related impairments in sensorimotor gating have been observed with AD progression, as well as in patients with psychosis [51,52,53,54,55]. Further, various rodent models of AD and psychosis have also exhibited deficits in pre-pulse inhibition of startle (PPI) [54,56,57]; however, we observed no PPI deficits in FAB rats when compared to their respective controls (Figure 3B).

To evaluate spatial learning and memory, we performed the Y-maze spontaneous alternation assay (Figure 3C). FAB rats (*n* = 10 rats; 51.19 ± 3.38%) displayed a significant decrease in percent alternations when compared to control rats (*n* = 9 rats; 67.02 ± 6.13%). A decrease in performance on this task is indicative of working memory deficits (*t*-test; *t* = 2.324 with 17 df; *p* = 0.033), which is a consistent observation in studies on FAB rats and in other rodent models used to study AD [45,58].

Lastly, while deficits in social behaviors have been reported in rodent models of AD [59], we did not observe any changes in social interaction between the control (*n* = 7 rats) and FAB-treated rats (*n* = 10 rats; Figure 3D).

### 2.4. FAB Rats Exhibited a Significant Decrease in Parvalbumin Protein in the Ventral Hippocampus

Decreases in parvalbumin-expressing interneurons have been reported in rodent models of psychosis [43,48,57,60], as well as in patients with psychosis [61,62] and AD [37,63]. Herein, we quantified parvalbumin protein levels in the vHipp using Western blot. Interestingly, FAB rats (*n* = 5 rats; 1.94 ± 0.15 uncalibrated optical density) displayed a significant decrease in parvalbumin protein, specifically in the vHipp, when compared to the controls (*n* = 5 rats; 4.32 ± 0.80 uncalibrated optical density; Mann–Whitney rank sum test; *p* = 0.008; Figure 4).

## 3. Discussion

The number of Americans living with AD is climbing rapidly and is projected to continue to increase, barring the discovery of a cure or breakthrough treatment to prevent or slow the progression of the disease. Currently, there is no cure for AD, but certain available FDA-approved drugs may help individuals to cope with symptoms and increase the quality of life of patients and caregivers [64,65]. Psychotic symptoms, including delusions and hallucinations, will occur in approximately 50% of patients over the course of their illness [1,2]. Furthermore, poorer disease outcomes are associated with psychotic symptoms in individuals with AD [66]. Specifically, a rapid decline in cognitive function and greater cognitive impairments have been reported in AD patients with psychosis compared to those without [3,66]. The antipsychotics commonly used to treat psychosis cause severe adverse events in AD patients, including cardiovascular and metabolic abnormalities, stroke, increased risk of parkinsonism, pneumonia, and a two-fold increased risk of death [9,10,67,68,69]. Thus, there is a dire need to develop therapies for AD patients who exhibit symptoms of psychosis.

AD is an age-related progressive disorder which can be categorized into two forms: familial and sporadic. The genetic, or familial, form of AD is a result of mutations in specific genes, such as amyloid precursor protein (APP), presenilin1 (PSEN1), and presenilin 2 (PSEN2). On the other hand, the sporadic form of AD encompasses about 95% of cases, and its etiology has yet to be completely elucidated. Evidence has led to the belief that various genetic (Apolipoprotein E (APOE) gene mutations), epigenetic, and environmental factors contribute to the development of this sporadic form of AD [70,71,72,73,74]. Although AD is an inherently human disorder, animal models that display AD-like phenotypes are essential to further our understanding of the pathologies contributing to AD. Herein, we utilized the FAB rodent model, which models the sporadic form of AD. In this model, an intracerebroventricular cannula is used to deliver a solution containing the human form of Aβ_42_ (aggregating properties), ferrous sulfate (pro-oxidative agent), and buthionine sulfoximine (facilitator of oxidative stress). This was used over a 3–4-week period to mimic an AD phenotype in Long–Evans rats [45]. The histopathologies present in the brains of FAB rats were consistent with observations in humans, and included hyperphosphorylated Tau protein, amyloid deposits, neuronal loss, and gliosis [45]. Further, previous studies using the FAB model have reported impairments in spatial memory in the Morris water maze task [45], and in this study, we also report deficits in cognition as measured by spontaneous alternation in the Y-maze. Given that progressive cognitive decline is a hallmark of AD and that AD patients with psychosis exhibit more severe cognitive impairments [3,66], we evaluated whether FAB rats displayed any changes in psychosis-related pathologies.

Psychotic symptoms can be observed in multiple mental disorders. They can be managed by antipsychotics, which exert effects by blocking dopamine D2 receptors, suggesting that the contribution of dopamine signaling is important [75,76,77]. Indeed, alterations in dopamine system function are often associated with symptoms of psychosis [76,77,78,79]. One region demonstrated to modulate the activity of VTA dopamine neurons is the vHipp [19]; however, it should be noted that the vHipp does not directly project to VTA dopamine neurons, but, rather, indirectly modulates the activity of these neurons via a multi-synaptic pathway involving the nucleus accumbens and the ventral pallidum (Figure 5). Furthermore, surgical [80,81], pharmacological [82,83], and cell-based therapies specifically targeting the vHipp can restore dopamine system function and reverse behavioral deficits in rodent models used to study psychosis [84]. Aberrant vHipp activity is observed in rodent models used to study schizophrenia, of which psychosis is a major symptom [19,48,49,50]. Indeed, the hippocampus is also a major site of pathology in AD, where alterations in its structure and function can be observed [15]. Specifically, progressive hippocampal atrophy is present in the anterior regions (analogous to ventral regions in the rodent) and underlies impairments in episodic memory [22]. These observations are significant because aberrant activity in the hippocampus could contribute to cognitive deficits, as well as to changes in dopamine system function, which is associated with psychosis.

The functional alterations and aberrant vHipp activity in the FAB rats observed in this study provide evidence that they may be useful for the study of comorbid psychosis-like pathology associated with AD. Herein, we measured spontaneous oscillatory activity by local field potentials (LFPs) in the vHipp of anesthetized rats and found a significant decrease in the delta (0–4 Hz) and theta (4–8 Hz) frequency ranges. Neuronal oscillations serve to coordinate activity into patterns that enhance local information processing and facilitate neurotransmission between regions [85]. It is not surprising that changes in the delta and theta rhythms generated by brain networks were observed, as they have been implicated in memory consolidation and rely heavily on hippocampal network integrity [86,87]. Further, these observations are consistent with other models of AD that display alterations in hippocampal network oscillations [87]. Additionally, FAB rats displayed increased firing rates of putative pyramidal neurons in the vHipp, which is indicative of hyperactivity within this region. Indeed, alterations in hippocampal activity, such as hyperexcitability and alterations in glutamatergic signaling, are present both in individuals with AD and in rodent models used to study AD [34,88,89,90]. The degree of hippocampal overactivation has been measured in AD patients and correlated with a decline in memory [90]. Aberrant hippocampal activity is also a common observation in rodent models that display psychosis-related symptoms [19,41,44], as well as in patients with psychosis [91,92,93].

Alterations in GABAergic markers, specifically in subpopulations of interneurons such as those containing the calcium-binding protein PV, have been observed in post mortem studies of patients who exhibited psychosis [62,94] and those diagnosed with AD [37,95]. It should also be noted that this reduction in PV expression is also a consistent observation in a wide variety of rodent models used to study psychosis [40,48,60,81,96,97,98], as well as those used in the study of AD [99,100]. Here, we demonstrate a decrease in PV expression in the vHipp of FAB-treated rats when compared to age-matched controls. We posit that that the loss of PV interneuron function in the vHipp resulted in aberrant hippocampal activity and was sufficient to produce an increase in dopamine neuron population activity.

Hippocampal hyperactivity and alterations in the hippocampal network oscillations present in the FAB model likely contributed to the dopamine dysfunction reported in this study. Specifically, FAB rats displayed significantly increased dopamine neuron population activity with no changes in the average firing rate or bursting activity. This is not surprising, because previous studies have identified distinct brain regions that can differentially modulate dopamine activity states [19,101,102]. Specifically, these measures are regulated by inputs from the ventral pallidum and tegmental nuclei (the latero-dorsal tegmentum and pedunculopontine tegmentum; for a review, see [103]). Dopamine neuron activity states, whether inactive (hyperpolarized) or active (single-spike or burst fire), can be modulated by various intrinsic and extrinsic mechanisms [103,104,105,106]. Population activity, a third activity state of dopamine neurons, is reliant on the number of spontaneously active dopamine neurons. Under normal conditions, approximately 50% of dopamine neurons are not active in vivo due to GABAergic inputs from the ventral pallidum [102,107]. These silent dopamine neurons are important as they allow for gain of function in the system [108], whereby the magnitude of the dopamine signal in response to a stimulus can be manipulated by regulating the number of spontaneously active neurons. The gain of function is particularly important because it is believed to contribute to psychosis, and several different animal models have displayed sustained increases in the number of spontaneously active dopamine neurons without any observable changes in firing rate or burst firing [19,109]. Of relevance to this study is the fact that aberrant drive from the hippocampus regulates the number of VTA dopamine neurons that fire spontaneously in a variety of rodent models of psychosis [19,40,47]. We demonstrated that FAB rats displayed alterations in dopamine neuron activity that were consistent with those observed in rodent models of psychosis.

Herein, we report various dopamine-dependent behaviors which correlate with psychosis-related pathologies that can be measured in rodent models. Human patients with psychosis exhibit increased sensitivity to psychomotor stimulants, which is suggested to reflect the aberrant mesolimbic dopamine neurotransmission thought to underlie symptoms of psychosis [77]. In rodents, this is modeled by examining the locomotor-inducing effects of the NMDA-antagonist, MK-801. Increases in locomotor activity to levels higher than those of the controls are often observed in rodent models with psychosis-like pathologies [19,110,111,112]. We observed locomotor hyperactivity in response to both the novel environment and MK-801 administration in FAB rats. We have previously demonstrated, in other models, that the stimulant-induced increase in locomotor activity can be attributed to hippocampal function, while changes in baseline locomotion are unaffected [19]. Activation of the vHipp in normal rats can increase the behavioral response to psychomotor stimulants [113], while vHipp inactivation reverses this hyperactivity [19]; therefore, the increased locomotor response to MK-801 observed in FAB rats is likely associated with vHipp-induced enhancement of dopamine neuron population activity [19]. We also evaluated PPI, which is a putative measurement of psychotomimetic effects and sensorimotor gating [57,83,114,115,116], and social interaction. As expected, PPI was dependent on the pre-pulse intensity in control rats. Reduced levels of PPI and social interaction have been observed in other rodent models with psychosis-related pathologies [57,111]; however, FAB rats displayed PPI and social interaction levels similar to the control rats. While these outcomes are, indeed, dopamine-dependent measures, they are complex behaviors modulated by multiple systems [117,118,119,120,121]. It is possible that the duration or severity of pathology induced by FAB was not sufficient to induce deficits in these behaviors.

Given the significant hippocampal pathology in AD combined with comorbid psychosis, we posited that FAB rats would display aberrant hippocampal function and activity, leading to pathological increases in VTA dopamine neuron population activity, and deficits in behaviors related to AD and psychosis. Indeed, in this study, we demonstrated that FAB rats displayed increases in vHipp and dopamine activity, thus introducing a potential novel model for comorbid psychosis in AD. In addition to cognitive deficits, FAB rats also displayed elevated baseline locomotor activity and an augmented response to the systemic administration of MK-801. These data are consistent with observations made in models of psychosis-related pathologies. The FAB model has already been commercialized and used for drug discovery and development, and our study provides evidence that this model can also serve as a useful tool to further investigate AD with comorbid psychosis, allowing for the identification of novel targets in order to develop therapeutic interventions for this devastating disorder. Indeed, drugs targeting α5 GABA-A receptors are of great interest, and are highly expressed in the hippocampus; they have been shown to have pro-cognitive and antipsychotic-like effects in vivo [82,122,123,124]. The results reported herein suggest that such compounds may have therapeutic utility for psychosis present in elderly AD patients.

## 4. Materials and Methods

All experiments were performed in accordance with the guidelines outlined in the USPH Guide for the Care and Use of Laboratory Animals, and were approved by the Institutional Animal Care and the Use Committee of UT Health San Antonio and the U.S. Department of Veterans Affairs.

### 4.1. Osmotic Minipump Survival Surgeries

Male Long–Evans rats were obtained from Envigo (Indianapolis, IN, USA). Survival surgeries were performed in a semi-sterile environment under general anesthesia. Rats (~325–350 g) were anesthetized with Fluriso (2–5% isoflurane, USP oxygen flow at 1 L/min) and placed in a stereotaxic apparatus using blunt, atraumatic ear bars. Osmotic minipumps were filled (control: Dulbecco’s phosphate-buffered saline (DPBS), no calcium, no magnesium; FAB: 1 µmol/L Aβ_42_, 1 mmol/L FeSO_4_, 12 mmol/L L-buthionine (S,R)-sulfoximine (BSO), and pH 5.1, brought to final volume with DPBS, no calcium, and no magnesium, as previously detailed) [45]. Pumps were primed via intracranial cannula directly into the lateral ventricle for a period of 24 h before delivery (A/P −0.9 mm and M/L +1.4 mm from Bregma; D/V −4.2 mm ventral of the brain surface). The cannula was fixed in place using dental acrylic and stainless-steel anchor screws, while the minipump was implanted subcutaneously into the back (slightly posterior to the scapulae). The FAB mixture was infused at a rate of 2.5 µL/hour for a period of three weeks prior to behavioral testing and electrophysiological recordings. The experimental timeline is depicted in Figure 6A.

### 4.2. In Vivo Electrophysiology and Local Field Potentials (LFPs)

For non-survival surgeries, rats were anesthetized with chloral hydrate (400 mg/kg; i.p.) prior to placement in a stereotaxic apparatus (Kopf; Tujunga, CA, USA). This anesthetic is required for dopamine neuron physiology as it does not significantly alter dopamine activity when compared to recordings in freely moving animals [115,125]. A core body temperature of 37 °C was maintained using a thermostatically controlled heating pad (PhysioSuite^®^; Kent Scientific, Torrington, CT, USA), while supplemental anesthesia was administered as required to maintain suppression of the limb withdrawal reflex. Extracellular glass microelectrodes (impedance 6–10 MΩ) were lowered into the vHipp (A/P −5.3 mm and M/L ±5.0 mm from Bregma; D/V −4.0 to −8.5 mm ventral of the brain surface) or VTA (A/P −5.3 mm and M/L ± 0.6 mm from Bregma; D/V −6.5 to −9.0 mm ventral of the brain surface). The firing frequency of spontaneously active putative pyramidal neurons in the vHipp was measured and identified as in previously published studies (neurons with firing frequencies less than 2 Hz [40,41,46]). LFP oscillatory activity was generated from the vHipp using open filter settings (low frequency cutoff: 0.3 Hz; high frequency cutoff: 300 Hz) and recorded for a minimum of ten minutes at a sampling rate of 10 kHz. Power in specific frequency bands was quantified with commercially available computer software (LabChart, Version 8; ADInstruments, Colorado Springs, CO, USA) using a standard Fourier transformation (Hamming window; 8 K). Spontaneously active dopamine neurons were identified using open filter settings (low frequency cutoff: 30 Hz; high frequency cutoff: 30 kHz) and previously established criteria [104,126]: (1) action potential duration >2 ms from the start to the end of the action potential; and (2) frequency between 0.5 and 15 Hz. Five to nine dorsal–ventral passes were made throughout the VTA, in a predetermined pattern (separated by 200 µm), to the medial and lateral regions. Dopamine cells were recorded for 2–3 min and three parameters of dopamine activity were measured: (1) population activity (the number of spontaneously active dopamine neurons encountered per track); (2) basal firing rate; and (3) the proportion of action potentials occurring in bursts (defined as the incidence of spikes with <80 ms between them; termination of a burst is defined by >160 ms between spikes). Electrophysiological recordings were analyzed by the Mann–Whitney rank sum test or two-way ANOVA followed by a Holm–Sidak post hoc test.

### 4.3. MK-801-Induced Locomotor Response

Rats were placed in an open-field arena (Med Associates, VT, USA), and spontaneous locomotor activity in the *x-y* plane was then determined by beam breaks and recorded with Open Field Activity software (Version 5, Med Associates). Following a 30-minute baseline recording, all rats were injected with MK-801 (0.75 µg/kg, i.p.) and recorded for an additional 45 min. The dose of MK-801 was specifically chosen to induce a modest locomotor effect in control rats, with the aim to observe an enhanced response in the FAB model. Locomotor data were analyzed by separate two-way ANOVAs, one for each of the relevant periods (baseline; 0.75 µg/kg MK-801), followed by a Holm–Sidak post hoc test.

### 4.4. Pre-Pulse Inhibition of Startle

Rats were placed in a sound-attenuated chamber (SD Instruments; San Diego, CA, USA) and allowed to acclimate to 65 dB background noise for 5 min prior to exposure to 10 startle-only trials (40 ms, 120 dB, and 15 s average inter-trial intervals (ITI)). Immediately afterward, rats were exposed to 24 trials during which a pre-pulse (20 ms at 69 dB, 73 dB, and 81 dB) was presented 100 ms before the startle pulse. Each pre-pulse + startle pulse was presented 6 times in a pseudo-random order (15 s average ITI). Startle responses were measured from 10 to 80 ms after the onset of the startle pulse and recorded using SR-Lab Analysis software (SD Instrument). PPI data were analyzed by two-way ANOVA, followed by a Holm–Sidak post hoc test.

### 4.5. Y-Maze Spontaneous Alternation Assay

The Y-maze consisted of three plastic arms separated by 120-degree angles. Each arm was 81 cm long and 20.32 cm wide, with walls 20 cm high. Rats were placed just inside the same arm, facing the center of the maze, and were allowed to move freely through the maze for a period of 10 min. During this time, the number and order of arm entries were recorded. Scoring consisted of recording each arm entry (defined as 80% of the rat’s body entering the arm). The percentage of alternation was then calculated as the number of alternations (triad containing entry to all three arms) divided by the total number of entries minus 2 and multiplied by 100. Y-maze data were analyzed by *t*-test.

### 4.6. Social Interaction (SI)

SI was assessed as previously described [127]. In brief, rats were acclimated to the testing arena (100 × 100 × 40 cm) individually for 10 min per day for 2 consecutive days prior to testing on day 3. On the test day, experimental rats were placed in the arena with a weight-matched “stimulus” rat. The 10 min test was scored by two blind experimenters, and the average time spent interacting (i.e., sniffing, climbing on, following, grooming, or wrestling) was reported. SI data were analyzed by *t*-test.

### 4.7. Western Blot

The protein expression of PV was measured using Western blot. The vHipp was dissected from a subset of control and FAB rats (*n* = 5 per group) and homogenized in ice-cold buffer (750 mL) containing a protease inhibitor. Samples were centrifuged (14,000 rpm for 2 min), and the supernatant, containing protein fractions, was collected. Protein concentrations were determined using the Bradford method prior to incubation with Laemmli sample buffer containing 5% dithiothreitol (10 min at 90 °C), then separated (20 µg/lane) at 200 mA on Any kD™ Mini-PROTEAN^®^ TGX™ Precast Protein Gels. Proteins were transferred (100 mA for 1 h) to a nitrocellulose membrane. The membranes were then washed 3 times (10 min each) in TBST and blocked (30 min in 5% BSA or milk in TBST) before incubation in rabbit anti-PV (1:5000; Abcam, Waltham, MA, USA) or mouse anti-GAPDH (1:1000; Abcam) antibody for 1 h. Membranes were washed 3 times (10 min in TBST), followed by either HRP-anti-rabbit 1:10,000 or HRP-anti-mouse 1:5000 (1 h at room temperature). Next, membranes were treated with Pierce™ ECL Western Blotting substrate (1 min) and exposed to high-performance chemiluminescence film (Amersham Hyperfilm™ ECL). Western blot films were scanned, and optical density was measured using ImageJ and analyzed by the Mann–Whitney rank sum test.

### 4.8. Histology

Upon the cessation of all electrophysiological experiments, rats were transcardially perfused (150 mL saline followed by 150 mL 4% formaldehyde in phosphate buffer). Brains were then extracted, post-fixed for 24 h, and cryoprotected (30% sucrose in 0.1 M phosphate buffer) for 72 h. To verify Fe^2+^ + Aβ_42_ + BSO brain infusion, silver staining was performed on a subset of brains that were sectioned coronally (50 µm) on a cryostat and processed using the instructions provided by the FD NeuroSilver Kit II. A separate subset of brains was sectioned (25 µm coronal sections) and processed with a Nissl stain for histochemical verification of cannula tracks into the lateral ventricles (Figure 6B). All brain sections were mounted on gelatin-coated slides and cover-slipped with DPX mountant prior to viewing on a Zeiss Axio Lab.A1 microscope (White Plains, NY, USA). Silver-stained brain slices were imaged with a dark field condenser (Figure 6C). Representative images were captured with a Zeiss AxioCam ICc 1 microscope camera, and all histology was performed with reference to a stereotaxic atlas [128].

### 4.9. Analysis

Electrophysiological analysis of vHipp and dopamine neuron activity was performed with commercially available computer software (LabChart version 8; ADInstruments; Chalgrove, Oxfordshire, UK). Locomotor activity was collected with Activity Monitor software (MED Associates; St. Albans, VT, USA). PPI data were collected using SR-Lab™ Analysis software (SD Instruments; San Diego, CA, USA). The optical density of the Western blots was measured using ImageJ. All data were analyzed using Prism software (GraphPad Software Inc.; San Diego, CA, USA). Data are represented as mean ± SEM, with *n* values representing the number of animals per experimental group, unless otherwise stated. Statistics were calculated using SigmaPlot (Systat Software Inc.; Chicago, IL, USA) and plotted with Prism software (GraphPad Software Inc.; San Diego, CA, USA). Significance was determined at *p* < 0.05.

### 4.10. Materials

Isoflurane (Fluriso™) was purchased from MWI Animal Health (Boise, ID, USA). Osmotic pumps (Alzet^®^, Model 2ML4, 0000327), Restore™ Western blot stripper (21059), Pierce™ ECL Western Blotting substrate (32106), and brain infusion kits (Alzet^®^, 0008663) were purchased from Thermo Fisher Scientific (USA). Chloral hydrate (C8383); DPX mountant (06522); iron (II) sulfate heptahydrate (215422); cOmplete™, Mini, EDTA-free Protease Inhibitor Cocktail (11836170001); anti-mouse IgG (whole molecule) peroxidase antibody produced in goat (A4416); and DPBS, but no calcium and no magnesium (14190144), were sourced from Sigma-Aldrich (St. Louis, MO, USA). Amyloid-β (1-42) peptide (20574) and L-buthionine-(S,R)-sulfoximine (14484) were purchased from Cayman Chemical Company (Ann Arbor, MI, USA). kD™ Mini-PROTEAN^®^ TGX™ Precast Protein Gels (#4569035), 2X Laemmli sample buffer (161-0737), and nitrocellulose/filter paper sandwiches (1620213) were purchased from BioRad (Hercules, CA, USA). Anti-parvalbumin antibody (ab114227), anti-GAPDH antibody (ab9484), and goat anti-rabbit IgG H&L (HRP) (ab6721) were purchased from Abcam (Waltham, MA, USA). An FD NeuroSilver Kit II was purchased from FD NeuroTechnologies, Inc. (Cat. # PK301A; Columbia, MD, USA). All other chemicals or reagents were of either analytical or laboratory grade, and were purchased from standard suppliers.

## Figures and Tables

**Figure 1 ijms-24-07196-f001:**
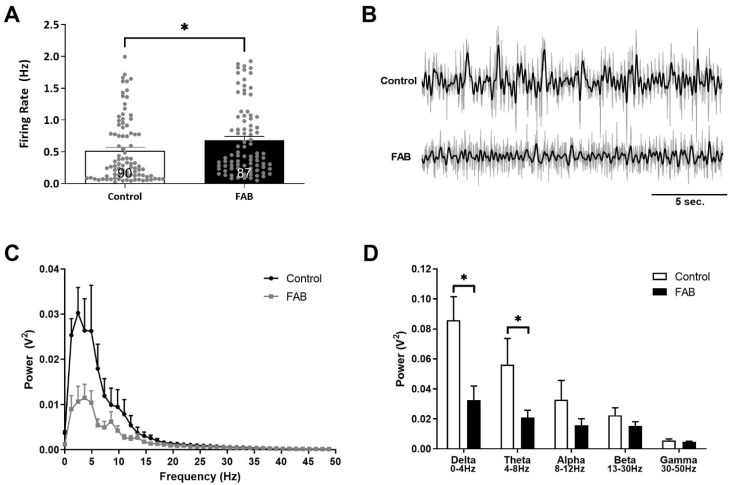
The average firing rate of spontaneously active putative pyramidal neurons in the ventral hippocampus (vHipp) was significantly higher in FAB rats, indicative of regional hyperactivity (**A**). Representative trace of spontaneous local field potential oscillations (filtered for <4 Hz; dark line) throughout the vHipp of a control (top) and FAB (bottom) rat (**B**). FAB rats displayed significant decreases in power in lower-frequency oscillations (**C**), specifically in the delta and theta frequency bands, as summarized in (**D**). * *p* < 0.05.

**Figure 2 ijms-24-07196-f002:**
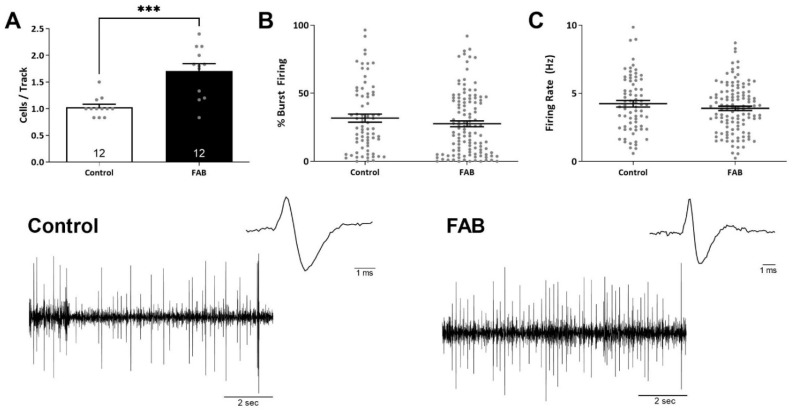
FAB rats exhibited increased ventral tegmental area (VTA) dopamine neuron population activity (**A**). No changes were observed in the percent burst firing (**B**) or the average firing rates (**C**) of dopamine neurons. Representative dopamine neuron trace and action potential from a control (left) and FAB (right) rat. *** *p* = 0.001.

**Figure 3 ijms-24-07196-f003:**
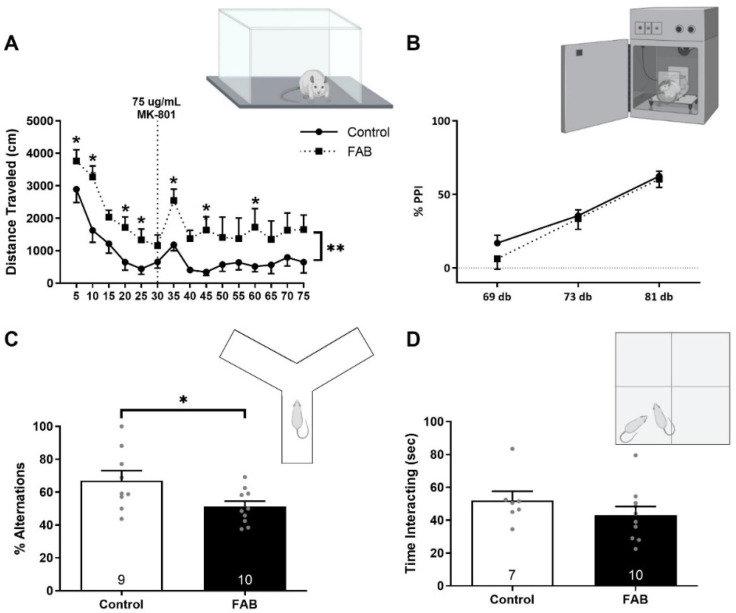
FAB rats displayed a baseline locomotor activity level higher than that of controls, which persisted after systemic MK-801 administration (**A**). No differences were observed in pre-pulse inhibition of startle (PPI) (**B**). A significant decrease in the percentage of spontaneous alternations in the Y-maze was observed in FAB rats (**C**). No differences were observed in the time spent interacting socially (**D**). * *p* = 0.033. ** denotes a main effect of strain *p* < 0.001. The icons used were adapted from BioRender.com (2021), retrieved from https://app.biorender.com/illustrations (accessed on 18 August 2021).

**Figure 4 ijms-24-07196-f004:**
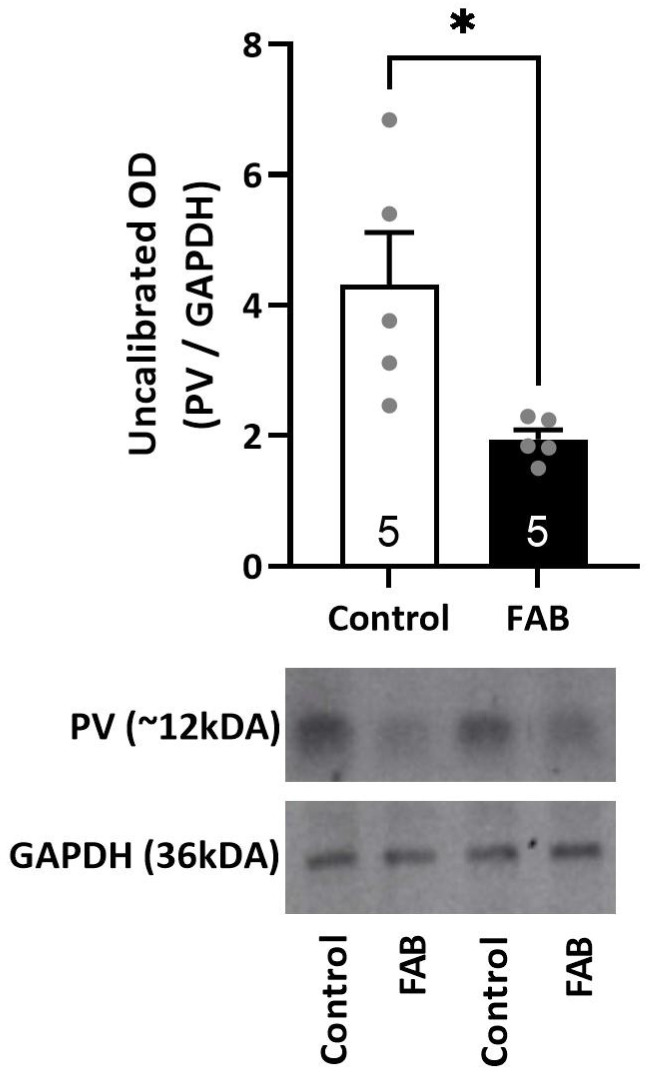
FAB rats displayed decreased levels of ventral hippocampal (vHipp) parvalbumin (PV). Representative Western blot films demonstrating PV and GAPDH levels in control and FAB rats. * *p* = 0.008.

**Figure 5 ijms-24-07196-f005:**
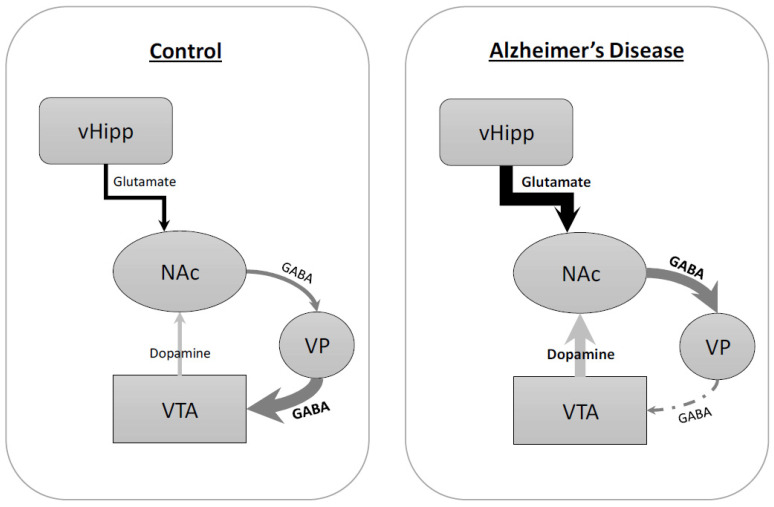
The ventral hippocampus (vHipp) can modulate activity in the ventral tegmental area (VTA) via a multi-synaptic pathway that includes the nucleus accumbens (NAc) and ventral pallidum (VP, **left**). In the case of psychosis, hyperactivity in hippocampal subfields can drive aberrant dopamine neuron activity by causing a disinhibition of the VTA by the VP, therefore modulating dopamine neurotransmission (**right**).

**Figure 6 ijms-24-07196-f006:**
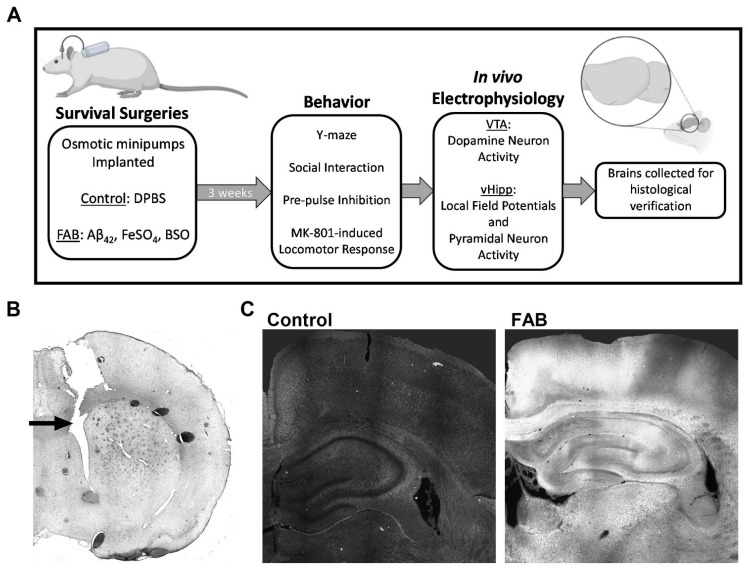
Schematic representation of the experimental timeline (**A**). Representative coronal brain slice displaying a cannula track into the lateral ventricles, indicated by a black arrow (**B**). Representative dark field image of a silver-stained brain from a control (left) and FAB (right) rat, with the brighter areas depicting areas of pathological deposits (**C**). 40× magnification. The icons used were adapted from BioRender.com (2021), retrieved from https://app.biorender.com/illustrations (accessed on 18 August 2021).

## Data Availability

All data collected for this study are reported herein and can be found associated with this manuscript.

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
