# Peer review of "Aberrant Dopamine System Function in the Ferrous Amyloid Buthionine (FAB) Rat Model of Alzheimer’s Disease"

_ijms, 2023, doi:10.3390/ijms24087196_

Round 1

Reviewer 1 Report

 Psychosis therapy during Alzheimer disease (AD) in elderly and old people is the important problem of molecular medicine. Stephanie M. Perez et al suggested that aberrant hippocampal activity observed in AD may contribute to dopamine dependent psychosis. Authors proved that the FAB model of AD may be useful for the investigation of comorbid psychosis related to AD and can help to create new antipsychosis drugs. The article is written in a good literary language, beautifully illustrated with graphs and microphotographs. Attention is drawn to a wide range of histological, biochemical and molecular research methods used in the work. At the same time, I would like to make some minor recommendations to improve the quality of the article.

1.       Authors provide information about psychosis developing during AD in the introduction (line 30-57). However, authors analyzed rather old literary sources, 1984-2014. It can recommend to analyze modern articles (2020-2022) on this issue from PubMed. For example:

https://pubmed.ncbi.nlm.nih.gov/33048274/

https://pubmed.ncbi.nlm.nih.gov/34983978/

https://pubmed.ncbi.nlm.nih.gov/34918337/

2.       Authors supposed that psychosis developing during AD can correlate with contain amyloid β (Aβ) in hippocampus (lines 58-100). However, authors analyzed in most causes old literary sources. Could authors find more news literature, 2019-2022? For example - https://pubmed.ncbi.nlm.nih.gov/33165946. This will emphasize the relevance of the investigation.

 3.       In accordance with the rules of the IJMS references in the text must be numbered in order of appearance in the text (including table captions and figure legends). Reference numbers should be placed in square brackets [ ], and placed before the punctuation; for example [1], [1–3] or [1,3]. https://www.mdpi.com/journal/ijms/instructions#preparation. Authors gave references in the text as a surnames. Please, correct.

 4.       Authors describes the materials and methods (4.1. Osmotic Minipump Survival Surgeries, 4.8. Histology) and refer to Figure 6. Unfortunately, I can’t find this figure in the article text. Please add it to the text of the article.

 5.       The Figure 1 is in the page 3 and in the page 12. Is it mistake?

Reviewer 2 Report

This paper hypothesizes that aberrant regulation of dopamine signaling in the hippocampus region of the brain is responsible for psychosis associated with Alzheimer’s disease (AD). They have used an established model of sporadic AD model, a FAB rat model system. They checked the activity of pyramidal neurons in the ventral hippocampus (vHipp) and observed hyperactivity along with functional hippocampal alterations. FAB rats also showed an increased dopamine population of neurons without any changes in neuron firing. This paper speculates the role of dopamine signaling from vHipp in causing psychosis associated with AD, which is important for understanding the occurrence and targeted treatment of psychosis symptoms in AD patients. I recommend the publication of this paper. However, the authors need to address the following points before it can be accepted for publication.

1.      The full form of VTA neurons should be mentioned on page 2, where it was introduced first. It is mentioned much later in the figure legend of fig 5. Similarly, the full form of PPI should be mentioned in line 148. In contrast, the authors have used the full form of FAB repetitively throughout the paper, which is not needed.

2.      In Fig 1D, what is the p-value of Alpha waves? The bar seems to have non-overlapping error bars.

3.      In Fig 2A, does Y-axis indicate the percentage of cells or the real cell number? If it is real, then there is an increase of only one cell in FAB rats. In figure legend, it is mentioned as VTA dopamine neuron population“ activity”. Shouldn’t it be density and not activity as there is no change in the firing of neurons? What could be the reason for not having increased neuronal firing despite increasing in numbers? if all those dopamine neurons are silent, then what can be the purpose of their increase?

4.      In Fig 3A, the difference between control and FAB rats, both with and without MK-801 seems the same. It is not convincing that MK-801 is stimulating locomotion in FAB rats. Did the authors try different doses of MK-801 to cause a more prominent effect or can they use some other NMDA antagonist, which is known to have a similar effect?

5.      References should be cited number-wise. It is very difficult to follow the references when readers need to search for each reference with the author's name.

6.      Line 228-229: “ It should be noted…..rather modulates used to study psychosis”. This sentence does not seem to convey the information properly.

7.      Line 239: The sentence starts with “ their activity via……..” seems incomplete.

8.      I think the authors should have done proper proofreading of the paper before submission. There are plenty of grammatical errors throughout the paper. Grammar mistakes including spelling mistakes, use of wrong articles or missing articles, inappropriate spacing, and wrong usage of singular-plural were observed in lines 90, 137, 143, 155, 164, 209, 248, 260, 268, 279, 326, 386, 441, 465, 477, 492, 508, etc. The spelling of “Cannula” is wrong throughout the paper except in the material and methods section. 

Round 2

Reviewer 2 Report

I am satisfied with the changes the authors have made to the manuscript and recommend the publication of this manuscript.